# Biomarkers Facilitate the Assessment of Prognosis in Critically Ill Patients with Primary Brain Injury: A Cohort Study

**DOI:** 10.3390/ijerph17124458

**Published:** 2020-06-21

**Authors:** Izabela Duda, Agnieszka Wiórek, Łukasz J. Krzych

**Affiliations:** Department of Anaesthesiology and Intensive Care, Faculty of Medical Sciences in Katowice, Medical University of Silesia, 14 Medyków Street, 40-752 Katowice, Poland; agnieszka.wiorek@med.sum.edu.pl (A.W.); lkrzych@sum.edu.pl (Ł.J.K.)

**Keywords:** S100 calcium-binding protein B, neutrophil gelatinase-associated lipocalin, neuron-specific enolase, matrix metalloproteinase 9, tissue inhibitor of metalloproteinase 1, primary brain injury, outcome prediction

## Abstract

Primary injuries to the brain are common causes of hospitalization of patients in intensive care units (ICU). The Acute Physiology and Chronic Health Evaluation (APACHE) II scoring system is widely used for prognostication among critically ill subjects. Biomarkers help to monitor the severity of neurological status. This study aimed to identify the best biomarker, along with APACHE II score, in mortality prediction among patients admitted to the ICU with the primary brain injury. This cohort study covered 58 patients. APACHE II scores were assessed 24 h post ICU admission. The concentrations of six biomarkers were determined, including the C-reactive protein (CRP), the S100 calcium-binding protein B (S100B), neuron-specific enolase (NSE), neutrophil gelatinase-associated lipocalin (NGAL), matrix metalloproteinase 9 (MMP-9), and tissue inhibitor of metalloproteinase 1 (TIMP-1), using commercially available ELISA kits. The biomarkers were specifically chosen for this study due to their established connection to the pathophysiology of brain injury. In-hospital mortality was the outcome. Median APACHE II was 18 (IQR 13–22). Mortality reached 40%. Median concentrations of the CRP, NGAL, S100B, and NSE were significantly higher in deceased patients. S100B (AUC = 0.854), NGAL (AUC = 0.833), NSE (AUC = 0.777), and APACHE II (AUC = 0.766) were the best independent predictors of mortality. Combination of APACHE II with S100B, NSE, NGAL, and CRP increased the diagnostic accuracy of mortality prediction. MMP and TIMP-1 were impractical in prognostication, even after adjustment for APACHE II score. S100B protein and NSE seem to be the best predictors of compromised outcome among critically ill patients with primary brain injuries and should be assessed along with the APACHE II calculation after ICU admission.

## 1. Introduction

Neurological diseases are an important cause of disability and mortality. Data from The Global Burden of Diseases, Injuries and Risk Factors study were used to estimate morbidity, disability, and mortality due to neurological diseases in the years 1990–2016. In 2016, neurological diseases were the second highest cause of death, accounting for 16.5% of all deaths [1]. Dysfunction of the central nervous system is common among patients in intensive care units (ICUs). It results from both primary and secondary damage to the brain [2]. The mechanisms of brain injury are complex and omnidirectional, but they unavoidably promote neuronal hypoxia, regardless of the underlying cause (e.g., traumatic brain injury or stroke). The response of neurons to damage includes the rapid onset of neuritic swellings, mitochondrial dysfunction, reduction of ATP concentration, the collapse of microtubules, and inactivation of neurons [3].

Clinical diagnostics of patients with central nervous system (CNS) damage is mainly based on computed tomography due to its high accuracy and accessibility [4]. Other invasive and non-invasive CNS monitoring methods have their limitations, especially in the ICU setting [5]. Common laboratory tests may also be used for basic monitoring. Among them is the relationship between the severity of the neurological condition and such parameters as sodium and glucose levels [6,7].

Biomarker determination is another method of monitoring the severity of a neurological condition, which has gained in popularity in recent years. The use of biomarkers to identify focal and diffuse brain lesions or potentially to diagnose intracranial pressure could be extremely valuable. An ideal biomarker should be characterized by high sensitivity and specificity. Unfortunately, it has been shown that there is no single marker that can predict the dynamics of changes in the neurological state with perfect diagnostic accuracy. The sensitivity and specificity of each analyzed biomarker separately are usually low, however, they increase significantly when several biomarkers are combined. Therefore, a panel of various indicators of CNS damage is usually required [8,9] in patients with traumatic brain injury (TBI), subarachnoid hemorrhage (SAH), or stroke [10,11,12,13] for assessment of prognosis or efficacy of therapy. 

Neuro-biomarkers can be assessed in cerebrospinal fluid and blood. Impaired blood–brain barrier integrity after injury causes biomarkers to enter the blood and can be identified by immunoassays. Biomarkers can be classified according to their role in brain injuries: inflammation and activation; protein degradation; necrosis and apoptosis; cytoskeleton damage; functional alteration; and endothelial alteration [14].

The Acute Physiology and Chronic Health Evaluation (APACHE) scoring system is among the many tools proven useful in daily practice for describing ICU patients, especially their clinical status on ICU admission, and predicting the possible course and outcome of the ICU stay [15]. The APACHE scoring concept was first introduced in the 1980s by Knaus et al. [16]. Over the years, many variations have been developed for clinical utilization, including APACHE II, III, and IV scores, among which, the APACHE II remains the most generally used score worldwide in the diverse settings of ICUs [17]. Still, it is by no means a perfect instrument, and attempts have been made to further improve its diagnostic accuracy, e.g., by recording the values of additionally selected biomarkers [18].

Therefore, we aimed to identify the best biomarker, along with the APACHE II score, in mortality prediction among patients admitted to the ICU with primary brain injuries. We decided to test a battery of six biomarkers found in serum, each of them qualified for identifying different pathophysiological pathways of CNS injury (i.e., C-reactive protein (CRP), S100 calcium-binding protein B (S100B), neuron-specific enolase (NSE), neutrophil gelatinase-associated lipocalin (NGAL), matrix metalloproteinase 9 (MMP-9), and tissue inhibitor of metalloproteinase 1 (TIMP-1)). 

## 2. Materials and Methods 

In this prospective observational study, we included 58 critically ill patients with primary brain damage who were admitted consecutively to the 10-bed mixed intensive care unit (ICU) in a university hospital over a twelve-month period. 

The Ethics Committee approved the study protocol and waived the need for obtaining the informed consent for participation in the study from the included patients (KNW/0022/KB1/86/13). All patients’ data were obtained in accordance with the national regulations of personal data management. 

Patients were admitted to the ICU with the diagnosis of the primary central nervous system (CNS) injury concomitant with at least a single-organ failure. The main causes of the primary CNS injuries were traumatic brain injury (TBI, *n* = 15, 26%), subarachnoid hemorrhage (SAH, *n* = 23, 39%), cardiac arrest (*n* = 16, 28%) stroke (*n* = 1, 2%), tumor (*n* = 2, 3%), and status epilepticus (*n* = 1, 2%). The pre-set exclusion criteria were age under 18 years, pregnancy, and postpartum recovery (*n* = 0). No a priori power or sample calculations were performed. 

The baseline demographic and clinical characteristics, clinical presentation during hospitalization, and laboratory results were prospectively collected and reviewed after the study period. The Acute Physiology and Chronic Health Evaluation II (APACHE II) score was calculated based on the worst values recorded within 24 h post-admission. 

Immediately upon admission (up to 2 h), a blood sample was collected for standard laboratory tests. The following parameters were registered: serum creatinine (Cr), bilirubin, white blood cell count, serum lactate concentration, platelet count, hemoglobin, and C-reactive protein concentration.

A subsample for biomarkers determination was extracted from the blood sample withdrawn for standard laboratory tests. The blood sample was collected from all patients once, on admission, over the whole period of the study and sent to the laboratory within 15 min of collection. The sample collection process followed the same time regimen for all included patients. On arrival to the laboratory, the blood sample was immediately centrifuged at 3000× *g* for 10 min. The supernatant was then separated and stored at −80 °C until further analyzed. The laboratory collected the samples from all patients included within the twelve-month study period and analyzed them after the end of the study population recruitment. Both laboratory workers and investigators were unaware of the values during patients’ hospitalization. The panel of biomarkers selected for concentration recording included the S100 calcium-binding protein B (S100B), neuron-specific enolase (NSE), neutrophil gelatinase-associated lipocalin (NGAL), matrix metalloproteinase 9 (MMP-9), and tissue inhibitor of metalloproteinase 1 (TIMP-1). 

Commercial ELISA enzyme immunometric assay kit was used for the assessment of quantitative serum concentrations of selected biomarkers according to the manufacturer’s instructions (S100B, NGAL: Biovendor, Brno, Czech Republic; NSE: Fujirebio Diagnostic AB, Göteborg, Sweden; MMP-9 and TIMP-1: Cloud-Clone Corp, Katy, TX, USA). All measurements were performed from the single blood sample taken from each patient, and all measurements were validated as required by the ELISA protocol set of standards and quality control reagents. 

ICU mortality was considered the outcome. A STROBE Statement (strengthening the reporting of observational studies in epidemiology) was applied for appropriate data reporting.

Statistical analysis was performed using MedCalc v.18 software (MedCalc Software, Ostend, Belgium). Quantitative variables were depicted using medians and interquartile ranges (IQR, i.e., 25–75 pc). The Shapiro–Wilk test was used to verify their distributions. Qualitative variables were described with frequencies and percentages. Between-group differences for continuous variables with normal distribution were assessed with independent samples student t-test and continuous variables with non-normal distribution were assessed using the Kruskal–Wallis test. For categorical variables, the Chi-squared test was applied. Odds ratios (OR) with their 95% confidence intervals (CI) were also calculated to express the associations between high values of the parameters and mortality. The correlation was assessed using Spearman’s rank correlation coefficient. Receiver operating characteristic (ROC) curves were drawn and areas under the ROC curves (AUC) were calculated to determine the predictive value of studied parameters and the outcome. The ROC analysis was also performed to statistically assess the optimal cut-off points in outcome prediction. Then, the cut-off point values of the parameters were applied to distinguish categories of high (i.e., ≥ cut-off) and low (i.e., < cut-off) values of the biomarkers. Logistic regression was performed to verify observations from bivariate models, taking into account the predictive values of the six analyzed biomarkers adjusted for the inclusion of the APACHE II score. It was calculated individually for the APACHE II paired with each biomarker, respectively, and the outcome variable was mortality. AUCs were also calculated to assess the diagnostic accuracy of the final logistic regression equations.

All tests were two-sided. A *p* value <0.05 was considered significant. 

## 3. Results

The study group consisted of 58 patients (30 men, 52%). The median age was 60.5 years (IQR 43–67). The median length of the ICU stay was 8 days (IQR 4–17). The median score in the APACHE II system was 18 (IQR 13–22) points. Detailed characteristics of the study subgroups are depicted in Table 1. Mortality was 40% (*n* = 23).

The medians of analyzed biomarkers for the entire study population were CRP 79.32 (IQR 13.59–160.98), NGAL 85.94 (IQR 47.85–145.92), NSE 7.04 (IQR 3.40–10.39), S100B 0.038 (IQR 0–0.415), MMP-9 172.00 (IQR 93.00–274.20), and TIMP-1 223.60 (IQR 193.10–313.50), respectively, Figure 1, Figure 2, Figure 3, Figure 4, Figure 5 and Figure 6. 

The results of bivariate investigations performed within the subgroups divided in terms of mortality are depicted in Table 2. Apart from the MMP-9 and TIMP-1, the concentrations of the studied biomarkers were statistically significantly higher in deceased patients. This observation was also confirmed by subgroup analyses in patients with TBI, SAH, and post-cardiac arrest syndrome (Table 3).

All investigated parameters did not differ significantly in terms of gender and cause of admission and did not correlate significantly with age and duration of ICU stay (*p* > 0.05 for all, data not shown). 

Mortality prediction by APACHE II score and individual biomarkers for the entire study population are depicted in Table 4. APACHE II predicted the outcome with AUC = 0.766 (95% 0.637–0.868). It was confirmed that except for MMP-9 and TIMP-9, the remaining investigated biomarkers were good predictors of an undesirable outcome.

High CRP (determined by the cut-off value in the AUC analysis, i.e., ≥34.26) was found in 35 subjects, high NGAL (i.e., ≥88.98) was found in 17 patients, high NSE (i.e., ≥9.15) was found in 10 patients, high S100B (i.e., ≥0.415) was found in 9 patients. High CRP, high NGAL, high NSE, and high S100B were associated with increased mortality (Table 5). 

Significant correlations were observed between APACHE II and several neuromarkers (Table 6).

The combination of APACHE II and four of the six studied biomarkers significantly improved diagnostic accuracy of mortality prediction. MMP-9 and TIMP-1 remained impractical in prognostication (Table 7).

## 4. Discussion

This study attempted to identify the best biomarker that when combined with the APACHE II score may aid in predicting mortality risk among patients admitted to the ICU with a primary brain injury. We selected six biomarkers for analysis. Four of them (S100B, NSE, NGAL, and CRP) showed very good predictive properties in patients after CNS injury. The S100B protein was characterized by the best predictive value in outcome prognosis, also in combination with APACHE II. NSE was also of high importance.

S100B has probably been studied most often and we have the most detail of it as a biomarker of brain injury. It is a calcium-binding protein and is considered a marker of damage and death to astrocytes [19]. S100B can be secreted from a variety of cells such as heart muscle, skeletal muscle, and the adipose tissue. However, it is mainly secreted from the glial cells and can, therefore, be measured in serum after brain injury. The mean normal blood concentration of S100B is 0.05 ng/mL, and after severe craniocerebral trauma, it may increase up to 100 times [20]. This is in line with our observations, as in the present study the S100B concentration in the non-survivors group was 75 times higher compared to the survivors. Moreover, out of all assessed biomarkers, the S100B had the highest calculated relative risk, meaning that the high concentration of S100B was associated with increased mortality by more than 5.5-times. The correlation between the S100B concentration and computed tomography (CT) images in severe TBI was described for the first time by Raabe et al. in 1999 [21]. Since then, the role of S100B as a prognostic biomarker in TBI has been extensively discussed in numerous studies [22,23,24,25]. All these studies share the common conclusion about the S100B protein being useful in brain assessment after the TBI.

The utility of S100B has also been a matter of interest in the context of subarachnoid hemorrhage. In the study by Sanchez-Pena et al. S100B was a good predictor of outcome in patients affected by SAH [26]. Similar observations were published by other researchers [27,28,29]. In contrast, in an ischemic stroke, S100B was proven to be of limited value as a diagnostic marker, even though it showed some prognostic power [30]. In studies of patients after cardiac arrest, determination of S100B concentration on admission, i.e., within 8 h after successful cardiopulmonary resuscitation (CPR), was a good predictor of 24-hour mortality [31].

The neuron-specific enolase is considered as a good predictor of adverse prognosis after cardiac arrest [32]. NSE occurs mainly in neurons and neuroendocrine cells and is a marker of neuronal damage [33]. A sudden increase in serum NSE concentration has been reported after various types of neurological damage, such as TBI, ischemic stroke, and cerebral hemorrhage [9,34,35]. Studies tend to combine measurement of NSE and S100B concentrations to emphasize that, albeit slightly weaker in comparison, the NSE indeed demonstrates high diagnostic and prognostic value [29,36]. Our results revealed a similar trend.

We observed similar high sensitivity and specificity to that of the S100B for the neutrophil gelatinase-associated lipocalin. NGAL is not a specific marker of brain damage. It is a small 25 kDa molecule that is covalently bound to the neutrophilic gelatinase and is present at low concentrations in human tissues including kidneys, lungs, stomach, and colon.

Released by neutrophils upon activation, NGAL is regarded as a marker of bacterial infection and systemic inflammation. After brain trauma, NGAL sequesters iron, leading to damage of cerebral tissue [37]. The study by Chen revealed significantly increased NGAL serum concentrations in patients with adverse outcomes observed within 90 days after SAH [38]. Likewise, observations performed by Zhao in patients after TBI showed a positive correlation between high NGAL expression and the severity of the injury [39].

CRP is also not a specific marker of neurological damage. It is a marker of a systemic inflammatory response and is not produced by brain tissue cells. However, the inflammatory response induced by trauma and SAH is an important element in the pathophysiology of brain injury. Cell and tissue damage trigger the release of inflammatory mediators, such as cytokines, chemokines, cytotoxic proteases, and oxygen radicals. This neuroinflammatory environment causes secondary brain injury, cell death, further progress of tissue loss of function, and neurodegeneration [40,41]. Increased CRP concentration has been reported in both SAH and TBI [42,43].

Our study showed that the four biomarkers mentioned above are characterized by a good predictive quality that increased even further when adjusted for the APACHE II score, which is already a tool of approved value in prognostication based on the severity of the disease and is calculated within the first day post-admission in most ICUs in the world [15]. We consider it beneficial to complement the assessment of the APACHE II score with the additional measurement of at least one of the discussed biomarkers. In our study, it noticeably improved the predictability and accuracy of the prognosis provided by the APACHE II score, which may sometimes be overlooked on its own due to routinization. 

Matrix metalloproteinase 9 is an extracellular matrix protein released by neutrophils after secondary ischemia and in neurodegenerative disorders. Inhibition of MMP-9 activity occurs after the impact of a specific TIMP-1 inhibitor. MMP-9 plays a key role in the pathophysiology of acute brain injury in SAH because it disrupts the blood–brain barrier, increases brain edema, and leads to neural and vascular apoptosis [44]. In the acute phase of the ischemic stroke, there is a significant increase in MMP-9 expression in the ischemic area and a noticeable increase in peripheral blood concentration [45]. In the study by Lorente, MMP-9 and TIMP-1 concentrations were associated with 30-day mortality in patients after TBI [46].

In our study, we did not demonstrate the prognostic usefulness of MMP-9 and TIMP-1 based on a one-time examination on admission. Secondary cerebral ischemia may develop several days after the primary injury, prompting MMP-9 increase as a part of the inflammatory response. Elevated MMP-9 concentration immediately after injury is probably associated with a transient blood–brain barrier disruption. We suspect that the low specificity and specificity for MMP-9 and TIMP-1 in our study can be associated with the blood sample being taken for analysis in the phase of restoration of the blood–brain barrier integrity and reduction of its permeability in the process.

In addition to high sensitivity and specificity, the biomarkers examined in our study were characterized by strong mutual correlations. The strongest correlation was observed between NGAL and CRP. This confirms the important role of the inflammatory response to brain damage.

In our study, the ELISA was used for biomarkers determination. Double-sided sandwich ELISA with detection based on absorbance, fluorescence, or chemiluminescence still seems to be the most convenient and reliable system and the method chosen most often by research laboratories working in the field of TBIs [47]. In the ELISA method, the reaction takes place on a 96-well plate coated with antibody or antigen depending on the type of test. Therefore, it is performed after collecting the appropriate number of samples and used to assess the prognostic value of the analyzed parameters. Numerous reports confirm the predictive utility of biomarkers in brain injury. The need for biomarker testing emerges in diagnostics, as they can be particularly useful in making decisions about further proceedings with a particular patient. This requires rapid, sensitive immunochromatographic tests comparable to the ELISA. Several reports describe the rapid test used in stroke. This biomarker panel includes , for example, S100B, among others. The test takes about 15 min to complete. The authors encourage further research aimed at introducing rapid tests into clinical practice [13,48].

Biomarkers found in serum after the injury of CNS tissue has been extensively studied as factors beneficial in outcome prognostication after the event, often addressed as a hypoxic-ischemic brain injury (HIBI) [49]. The scale of HIBI observed in a patient most likely is reflected in the release and increase of serum concentrations of certain proteins, such as NSE and S100B [50]. Hypoxia- and ischemia-derived brain injury can be observed after an event such as cardiac arrest, stroke, cerebral infarction, a complication of traumatic brain injury, and delayed consequence of intracranial hemorrhage. Patients with such underlying causes of hospital admission are often assessed in the lower ranges of the prognostic scores such as Glasgow Coma Scale (GCS), and, in effect, often require a certain amount of sedation upon admission to the intensive care unit [49]. However, it is not uncommon to base the decisive points of the patient’s therapeutic plan on somewhat subjective standards of neurological assessment, and even standardized outcome prediction tools and scores allow for a certain freedom in individual assessment. Therefore, the suspicion of possible poor outcome and resulting decisions to implement life-sustaining treatment or withdraw from further invasive therapy should not be based solely on physical examination. The biomarkers provide an objective indicator, as their serum concentrations are presumably not affected by applied drugs, such as sedatives, analgesics, or neuromuscular relaxants that affect the patient’s general neurologic state [49]. 

### Study Limitations

To the best of our knowledge, the presented study is the first report analyzing a panel of biomarkers in critically ill patients admitted to the ICU due to brain injury of various etiologies. However, one should bear in mind possible limitations regarding the presented study. Firstly, it is observational in design, and all medical interventions undertaken were not influenced by the study protocol but based on the best knowledge and decisions of the attending physicians. In addition, this is a single-center observation with a relatively small study group, so extrapolation of our results is limited. No power analysis was performed to assess the sample size, so one ought to realize the potential selection bias. However, as biomarker measurements are not yet commonly performed in our hospital, we included all suitable cases of patients with results of the complete panel of parameters. Due to the heterogeneous nature of the study population, various principal and additional medical diagnoses could affect the results. However, heterogeneous clinical conditions are characteristic of the mixed ICU patient population. Our results reflect the scenario of our actual daily clinical practice. 

The number of cases in our study is relatively limited. However, there is a lack of statistically significant influence of the cause of admission on the prognostic value of evaluated biomarkers. This fact lets us eliminate the cause of admission as a confounding factor and allowed for the utilization of selected biomarkers for prognostication of patients admitted to the ICU due to the primary brain injury regardless of the mechanisms of its occurrence. The strength of the study comes also from the fact that there is a paucity of studies comparing many biomarkers in terms of their value in outcome prediction and assessing their interdependence. Studying mutual relations between serum biomarker concentrations may aid in the future development of combined marker panels for daily practice, with optimal usefulness in determining the best therapeutic steps, however, the issue requires further research. 

Conclusions drawn from lab tests’ results should always be analyzed with consideration of the physical examination and comprehensive clinical assessment. Therefore, we encourage the treatment of biomarkers as useful tools for objectifying the decision-making processes and as a valuable supplements to the goal-directed, individualized approach to the patient, in addition to the universally recommended guidelines. 

## 5. Conclusions

We have found that S100B, NSE, NGAL, and CRP seem to be of high quality in predicting mortality following primary brain injury of various etiologies. The S100B protein and NSE appear, however, to be the best predictors of the compromised outcome among critically ill patients with primary brain injury, which should be assessed along with APACHE II calculation after ICU admission. The combination of information on alterations in homeostasis using a simple scoring system and different biomarkers of distinct cellular origin and reflecting specific pathophysiological processes may considerably improve prognostication. 

## Figures and Tables

**Figure 1 ijerph-17-04458-f001:**
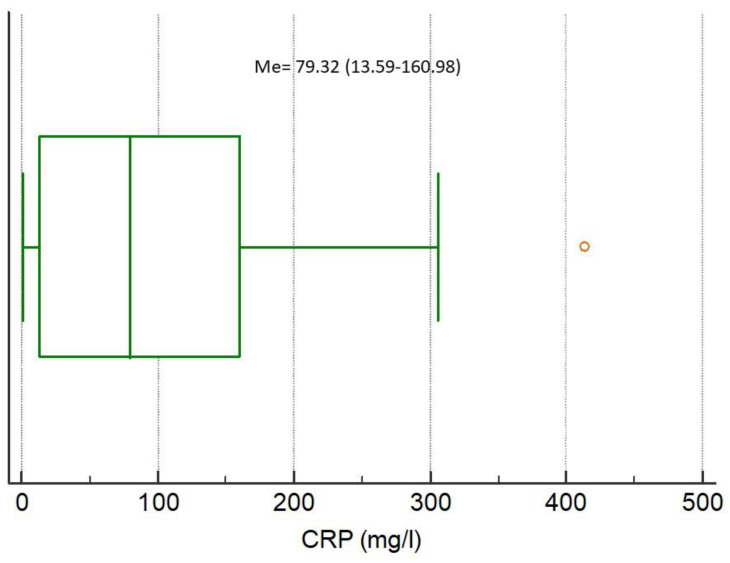
Distribution of the C-reactive protein (CRP) concentrations in the study population. Circles indicate outliers and far out values illustrated as single data points.

**Figure 2 ijerph-17-04458-f002:**
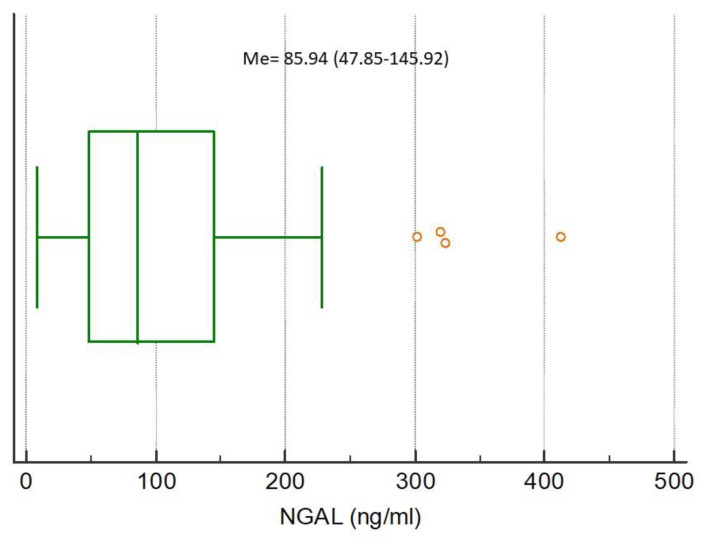
Distribution of the neutrophil gelatinase-associated lipocalin (NGAL) concentrations in the study population. Circles indicate outliers and far out values illustrated as single data points.

**Figure 3 ijerph-17-04458-f003:**
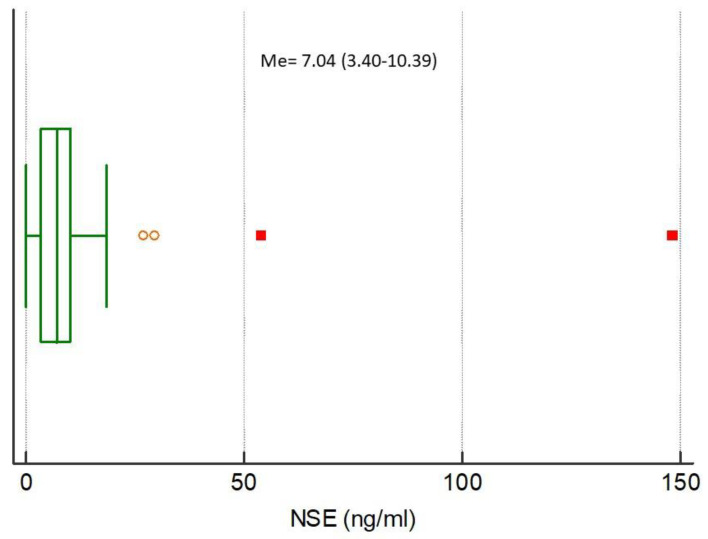
Distribution of the neuron-specific enolase (NSE) concentrations in the study population. Circles and squares indicate outliers and far out values illustrated as single data points.

**Figure 4 ijerph-17-04458-f004:**
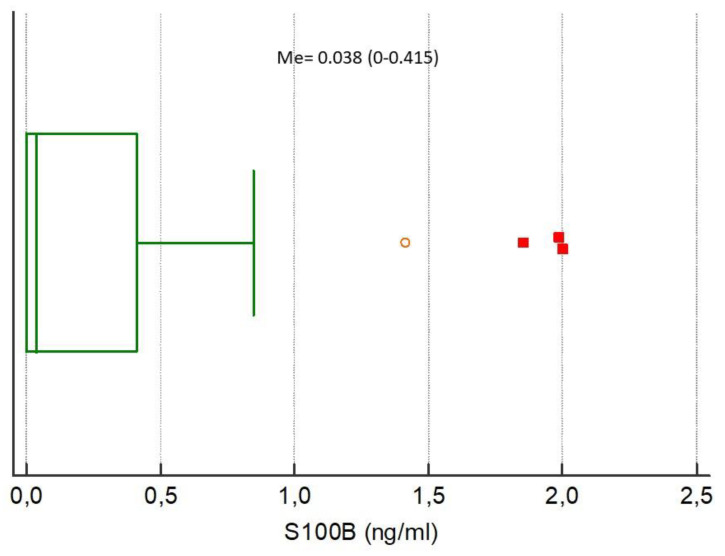
Distribution of the S100 calcium-binding protein B (S100B) concentrations in the study population. Circles and squares indicate outliers and far out values illustrated as single data points.

**Figure 5 ijerph-17-04458-f005:**
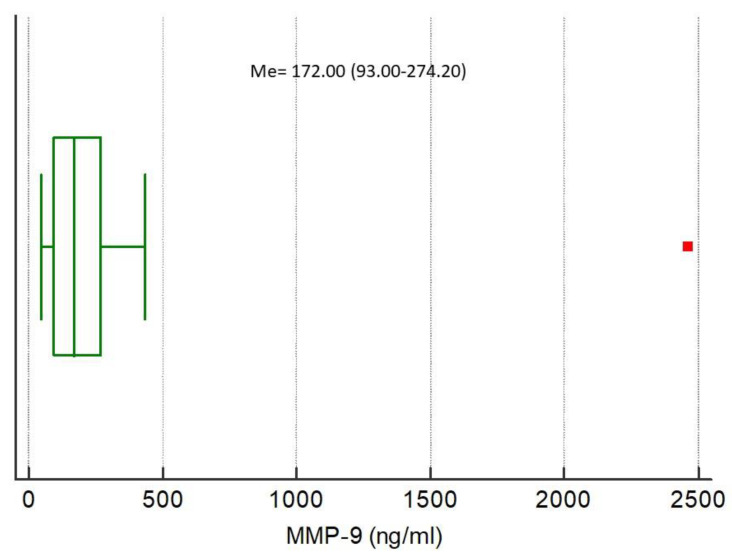
Distribution of the matrix metalloproteinase 9 (MMP-9) concentrations in the study population. Squares indicate outliers and far out values illustrated as single data points.

**Figure 6 ijerph-17-04458-f006:**
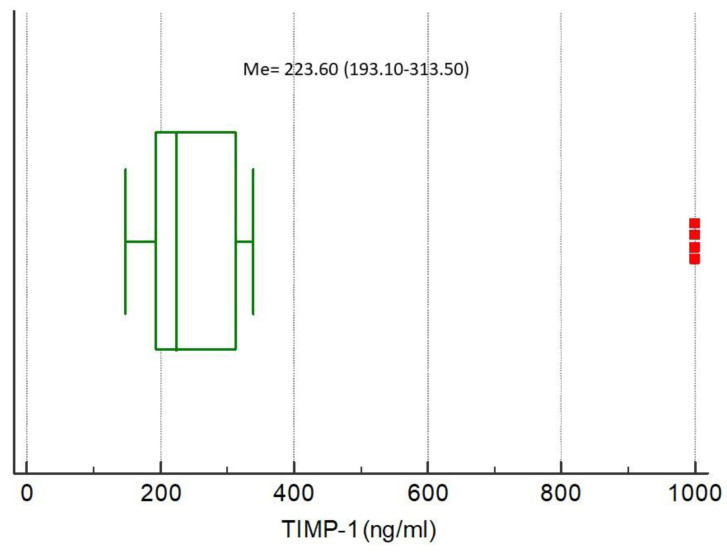
Distribution of the tissue inhibitor of metalloproteinase 1 (TIMP-1) concentrations in the study population. Squares indicate outliers and far out values illustrated as single data points.

**Table 1 ijerph-17-04458-t001:** Study group characteristics and procedure-related variables ^1^.

Variable	Unit	All (*n* = 58)	Survivors (*n* = 35)	Deceased (*n* = 23)	*p*
Sex	Males	30 (52%)	20 (57%)	10 (43%)	0.13
	Females	28 (48%)	15 (43%)	13 (57%)	
Age	(years)	61 (43–67)	50 (38–62)	66 (57–74)	<0.05
Cause of admission	Traumatic Brain Injury	15 (26%)	11 (31%)	4 (17%)	0.212
	Cardiac arrest	16 (28%)	6 (17%)	10 (44%)	
	Subarachnoid hemorrhage	23 (39%)	16 (46%)	7 (30%)	
	Tumor	2 (3%)	1 (3%)	1 (4.5%)	
	Stroke	1 (2%)	0	1 (4.5%)	
	Status epilepticus	1 (2%)	1 (3%)	0	
ICU length of stay	(days)	8 (4–17)	9 (4–17)	8 (3–13)	0.57
APACHE II score	(points)	18 (13–22)	15 (9.0–20.0)	22 (17–24)	<0.05
Creatinine	(mg/dL)	1.04 (0.82–1.37)	0.88 (0.79–1.21)	1.37 (1.08–2.33)	<0.001
Bilirubin	(mg/dL)	0.83 (0.49–1.29)	0.84 (0.62–1.15)	0.67 (0.42–1.43)	0.60
Lactates	(mmol/L)	2.20 (1.45–3.15)	1.90 (1.20–2.58	2.90 (1.60–4.03)	0.004
White blood cells	(×10^9^/L)	15.02 (12.81–17.24)	15.49 (11.61–19.38)	14.47 (11.99–16.95)	0.64
Hemoglobin	(g/dL)	11.10 (10.28–11.93)	10.95 (9.83–12.08)	11.28 (9.82–12.74)	0.696
Platelets	(×10^9^/L)	162 (136–187)	175 (144–207)	146 (98–193)	0.245

^1^ Values are medians, interquartile ranges (Q1–Q3) for quantitative variables with non-normal distribution, means and 95% confidence interval (CI) for quantitative variables with normal distribution, and frequencies and percentages for qualitative variables. APACHE II score—Acute Physiology and Chronic Health Evaluation II, ICU—intensive care unit, IQR—interquartile range.

**Table 2 ijerph-17-04458-t002:** Concentrations of studied biomarkers assessed in deceased patients and survivors ^1^.

Variable	Unit	Survivors (*n* = 35)	Deceased (*n* = 23)	*p*
CRP	(mg/L)	30.54 (6.76–131.63)	120.47 (66.00–207.82)	0.003
NGAL	(ng/mL)	60.78 (37.11–90.48)	126.47 (90.12–311.12)	0.002
NSE	(ng/mL)	6.83 (2.08–7.50)	11.07 (6.96–28.33)	0.009
S100B	(ng/mL)	0.008 (0–0.092)	0.584 (0.084–1.637)	<0.001
MMP-9	(ng/mL)	167.50 (93.80–241.10)	188.00 (88.80–306.00)	0.792
TIMP-1	(ng/mL)	208.55 (190.65–277.10)	305.05 (193.10–1000.00)	0.247

^1^ Values are medians and IQR. *p* values refer to differences in strata defined by consecutive variables (e.g., CRP, NGAL, etc.). CRP—C-reactive protein, NGAL—neutrophil gelatinase-associated lipocalin, NSE—neuron-specific enolase, S100B—S100 calcium-binding protein B, MMP-9—matrix metalloproteinase 9, TIMP-1—tissue inhibitor of metalloproteinase 1.

**Table 3 ijerph-17-04458-t003:** Concentrations of studied biomarkers assessed in deceased patients and survivors in subgroup analyses in patients with traumatic brain injury (TBI), subarachnoid hemorrhage (SAH), and post-cardiac arrest syndrome ^1^.

Variable	TBI (*n* = 15)			Cardiac Arrest (*n* = 16)			SAH (*n* = 23)		
	Survival *n* = 11	Death *n* = 4	*p*	Survival *n* = 6	Death *n* = 10	*p*	Survival *n* = 16	Death *n* = 7	*p*
APACHE II	17 (14–20)	22 (17–26)	0.213	12 (8–19)	23 (21–28)	0.009	13 (9–24)	18 (17–21)	0.241
CRP	75.64 (16.10–129.35)	97.21 (32.17–161.50)	0.361	21.25 (5.39–131.63)	105.96 (66.00–242.00)	0.159	20.00 (5.82–141.25)	148.50 (70.52–201.00)	0.052
NGAL	81.23 (41.39–127.85)	95.03 (90.81–99.24)	0.602	47.21 (32.85–120.98)	311.12 (228.33–323.43)	0.019	57.51 (37.11–82.89)	89.43 (70.60–101.96)	0.237
NSE	6.85 (2.74–8.23)	28.19 (2.39–53.98)	0.602	8.14 (3.68–12.53)	22.74 (10.39–29.52)	0.088	5.10 (0.69–7.01)	7.07 (7.06–10.57)	0.091
S100B	0.059 (0–0.128)	0.708 (0–1.416	0.686	0.021 (0–0.068)	0.995 (0.035–1.987)	0.087	0 (0–0.030)	0.726 (0.513–0.817)	0.008
MMP-9	94.60 (76.00–165.40)	109.30 (63.60–155.00)	0.564	322.20 (304.30–356.70)	306.00 (143.10–1921.95)	0.655	161.00 (93.00–208.00)	171.70 (84.10–241.50)	0.831
TIMP-1	234.00 (211.43–267.98)	190.65 (172.20–209.10)	0.248	657.00 (334.00–1000.00)	556.20 (485.13–1000.00	0.564	205.90 (182.70–274.90)	244.85 (170.50–318.05)	0.831

^1^ Values are medians and IQR. *p* values refer to differences in strata defined by consecutive variables (e.g., CRP, NGAL, etc.). APACHE II—Acute Physiology and Chronic Health Evolution II, CRP—C-reactive protein, NGAL—neutrophil gelatinase-associated lipocalin, NSE—neuron-specific enolase, S100B—S100 calcium-binding protein B, MMP-9—matrix metalloproteinase 9, TIMP-1—tissue inhibitor of metalloproteinase 1.

**Table 4 ijerph-17-04458-t004:** ROC curve analysis for mortality prediction by the investigated biomarkers ^1^.

Variable	Units	AUC (95% CI)	*p*	Cut-Off	Sensitivity	100-Specificity
APACHE II	(points)	0.766 (0.637–0.868)	<0.0001	>15	91.30	54.29
CRP	(mg/L)	0.733 (0.598–0.843)	0.0006	>34.26	86.36	55.88
NGAL	(ng/mL)	0.833 (0.666–0.939)	<0.0001	>88.98	83.33	72.73
NSE	(ng/mL)	0.777 (0.601–0.901)	0.002	>9.15	58.33	90.91
S100B	(ng/mL)	0.854 (0.691–0.951)	<0.0001	>0.415	66.67	100.00
MMP-9	(ng/mL)	0.533 (0.311–0.746)	0.807	>208	50.00	75.00
TIMP-1	(ng/mL)	0.646 (0.416–0.835)	0.274	>279.3	60.00	83.33

^1^ AUC-area under the ROC curve, 95% CI—95% confidence interval, APACHE II—Acute Physiology and Chronic Health Evolution II, CRP—C-reactive protein, NGAL—neutrophil gelatinase-associated lipocalin, NSE—neuron-specific enolase, S100B—S100 calcium-binding protein B, MMP-9—matrix metalloproteinase 9, TIMP-1—tissue inhibitor of metalloproteinase 1.

**Table 5 ijerph-17-04458-t005:** High concentrations of the investigated biomarkers and ICU mortality ^1^.

Variable	High Concentration	OR (95% CI)	*p*
CRP	≥34.26 mg/L	7.13 (1.77–28.65)	0.006
NGAL	≥88.98 ng/mL	10.71 (1.84–62.49)	0.008
NSE	≥9.15 ng/mL	8.87 (1.66–47.26)	0.011
S100B	≥0.415 ng/mL	42.00 (4.05–435.04)	0.002

^1^ OR—odds ratio, 95% CI—95% confidence interval, CRP—C-reactive protein, NGAL—neutrophil gelatinase-associated lipocalin, NSE—neuron-specific enolase, S100B—S100 calcium-binding protein B.

**Table 6 ijerph-17-04458-t006:** Correlations between APACHE II and the studied biomarkers ^1^.

Variable	APACHE II
CRP	R = 0.199 *p* = 0.142
NGAL	R = 0.383 *p* = 0.025
NSE	R = 0.252 *p* = 0.150
S100B	R = 0.470 *p* = 0.005
MMP-9	R = 0.249 *p* = 0.264
TIMP-1	R = 0.027 *p* = 0.907

^1^ Values are Spearman’s ranks of coefficients of correlation, APACHE II—Acute Physiology and Chronic Health Evolution II, CRP—C-reactive protein, NGAL—neutrophil gelatinase-associated lipocalin, NSE—neuron-specific enolase, S100B—S100 calcium-binding protein B, NS—non-significant.

**Table 7 ijerph-17-04458-t007:** Mortality prediction using a combination of APACHE II and the studied biomarkers ^1^.

Variable	AUC (95% CI)	*p*
APACHE II + S100B	0.947 (0.811–0.995)	<0.0001
APACHE II + NSE	0.909 (0.760–0.980)	<0.0001
APACHE II + NGAL	0.886 (0.731–0.969	<0.0001
APACHE II + CRP	0.821 (0.695–0.910)	<0.0001
APACHE II + TIMP-1	0.708 (0.478–0.880)	0.166
APACHE II + MMP-9	0.667 (0.436–0.850)	0.342

^1^ AUC—area under the ROC curve, 95% CI—95% confidence interval, APACHE II—Acute Physiology and Chronic Health Evolution II, CRP—C-reactive protein, NGAL—neutrophil gelatinase-associated lipocalin, NSE—neuron-specific enolase, S100B—S100 calcium-binding protein B, MMP-9—matrix metalloproteinase 9, TIMP-1—tissue inhibitor of metalloproteinase 1.

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
