# Peer review of "Biomarkers Facilitate the Assessment of Prognosis in Critically Ill Patients with Primary Brain Injury: A Cohort Study"

_ijerph, 2020, doi:10.3390/ijerph17124458_

Round 1

Reviewer 1 Report

Congratulations for a very good publication about biomarkers, there can help to monitor the severity of neurological status. Addition of biomarkers (S100B and NSE) to the APACHE II estimate seems very useful.

By estimating the outcome I suggest to estimate 30 and 90 days mortality in addition to the in-hospital. Does age have an influence on outcome, independent of other risk factors?

Author Response

Thank you kindly for your appreciative voice towards our work.

We would also like to thank you for your suggestion regarding the analysis of 30 and 90 days mortality in the studied population. This is a valid point and we will certainly consider it in our future projects when further studying and expanding upon the topic. Unfortunately, we do not have such data now so we are unable to reveal them.

Age was analyzed in our cohort as an independent variable. Deceased patients were older than survivors (Table 1). However, age did not predict mortality in bivariate models. Therefore there was no reason to include it in further analyses. More importantly, we did not include age as a separate predictor in AUROC analysis because this parameter is one of the constituents of APACHE II score and there is significant collinearity between these two variables. 

Kind regards, the Authors.

Reviewer 2 Report

This is an interesting study investigating biomarkers for brain injury. I have some reviews that I think need to be made so that the article is more clear. 

Abstract:

-Remove all statistics and describe you results

-Why where the biomarkers chosen? Or did they come up on a screen, this needs to be clarified.  

Materials and Methods:

-Was the timeline of collection of samples the same for living vs. non living people?

-I think the article would benefit from including an experimental timeline of all manipulations (e.g. collection of samples)

-The ethics statement should be included earlier in the study

Results:

-Were there any sex differences observed?

-Were all different brain injuries grouped together?

Discussion

-Remove all statistics from the discussion

-Did the authors find a neuronal bio maker that could used?

-I think that the conclusions need to be toned down in the paper, especially because of the sample size of the study and that several types of injury were grouped together. This should be revised in the abstract and study conclusion.

-Could the authors include a power analysis of their study? This is can easily be done and should be included.

Author Response

Thank you kindly for the detailed review of our study. We hope our responses will rise to your expectations regarding applied corrections.

Abstract:

  1. We removed some statistical data from the abstract. However we believe that description of AUROCs for biomarkers is necessary as it was the main goal of the study to investigate the diagnostic accuracy of the models we had created, and ROC analysis is considered to be the cornerstone in this type of investigations (lines: 21-25)
  2. We clarified the reason for choosing this specific set of biomarkers. (lines: 19-20)

Materials and Methods:

  1. We collected the samples from 58 consecutive patients admitted to the ICU over a twelve-month study period. The samples were taken upon ICU admission. The timeline of the collection of samples was the same for the deceased patients and the survivors. We clarified this in the Materials and Methods section. (lines: 80-81, 96-97, 102-109)
  2. We included an experimental timeline of manipulations as suggested. (lines: 96-97, 102-109)
  3. We included the ethics statement earlier in the study as requested. (lines: 82-85)

Results:

  1. No differences in terms of sex and the studied parameters were observed. (Table 1, lines: 197-198)
  2. We sought to find the universal biomarker that could be useful in prognostication in patients with primary brain injury, regardless of the initial cause of this injury. Therefore, the different causes of brain injury were analyzed collectively. The subgroup division was performed only in terms of mortality. (lines: 153, 179-180, 199-200)

Discussion:

  1. We removed all statistics from the discussion as requested. (lines: 233-235, 262-263, 270-271)
  2. We established that out of investigated biomarkers, the NSE (which location is primarily neuronal) and the S100B (which location is primarily glial), as well as the NGAL combined with the APACHE II score could be used in predicting mortality following primary brain injury of various etiologies, according to our results. (lines: 367-370)
  3. We toned down the study conclusions as suggested. (lines: 27, 367-373)
  4. Based on available studies regarding S100B in mortality prediction in TBI patients (ex. Rodriguez et al. 2012, Shakeri et al. 2015, Olivecrona et al. 2009) we decided to include a similar group of patients. However, no a priori power or sample calculation was performed. (lines: 90-91). We mentioned this drawback in study limitations (lines: 344-345).

Kind regards, the Authors.

Reviewer 3 Report

Duda et al provide a compelling dataset for the use of multiple biomarkers (several previously identified in the literature) to predict mortality in patients with primary brain injury. Overall, the quality of the data is fantastic, but the manuscript would benefit from some minor adjustments in presentation and clarification.

  • What, if any, test re-test controls were performed to ensure that the abundance of the biomarkers is consistent in the same individuals? (e.g. two vials of blood taken at same time and tested, two different aliquots from same vial, etc)
  • What, if any, blinding occurred during analysis so that the researchers were unbiased in their observations?
  • Please provide graphs with data point so the reader can see the population distribution (standard error and deviations do not give the full picture)
  • Also, please provide a data table for the patients with their outcome and abundance of each biomarker measured. It would be helpful to know is there is a subgroup of responders for each marker or if it is a majority and the table would provide more data to inform future studies.

Author Response

We appreciate your inquisitive review of our paper and your expressed appreciation of our results. Please find below our responses and corrections based on your provided suggestions.

  1. We took one vial of blood from each patient. The blood samples were centrifuged at 3000 g for 10 min. The serum supernatant was separated and stored at -80o Each sample was checked with provided by the manufacturers quality control reagents and standards to ensure that the acquired results are trustworthy. (lines: 102-109, 113-118)
  2. The laboratory diagnosticians and investigators were blinded when it came to the knowledge of which samples were taken from the patients who died later on during the ICU stay. We specified this in the Materials and Methods section. (lines: 108-109)
  3. We revealed basic data on biomarkers across the population (i.e. box-and-whiskers). (Figure 1-6)
  4. We provided additional data on biomarkers in relation to mortality. We focused only on 3 major subgroups that were most numerous and homogeneous (i.e. TBI, post-cardiac arrest and SAH). (Table 3).

Kind regards, the Authors.
